# Effect of Ketogenic Diets on Body Composition and Metabolic Parameters of Cancer Patients: A Systematic Review and Meta-Analysis

**DOI:** 10.3390/nu14194192

**Published:** 2022-10-08

**Authors:** Haobin Zhao, Han Jin, Junfang Xian, Zhifu Zhang, Junling Shi, Xiaosu Bai

**Affiliations:** 1Department of General Practice, People’s Hospital of Longhua, 38 Jinglong Jianshe Road, Shenzhen 518109, China; 2Key Laboratory for Space Bioscience and Biotechnology, School of Life Sciences, Northwestern Polytechnical University, Xi’an 710072, China

**Keywords:** ketogenic diets, body composition, metabolic parameters, cancer patients, food function, nutrition

## Abstract

A ketogenic diet characterized by high fat and low carbohydrate can drive the body to produce a large number of ketone bodies, altering human metabolism. Unlike normal cells, tumor cells have difficulty in consuming ketone bodies. Therefore, the application of ketogenic diets in cancer therapy is gaining attention. However, the effect of ketogenic diets on body parameters of cancer patients is not well established. This meta-analysis aimed to summarize the effects of ketogenic diets on cancer patients in earlier controlled trials. PubMed, Embase, and Cochrane Library were searched for clinical trials that enrolled cancer patients who received ketogenic diets intervention. Ten controlled trials were included in this meta-analysis. Data were extracted and checked by three authors independently. Pooled effect sizes revealed a significant effect of ketogenic diets on body weight (SMD −1.83, 95% CI −2.30 to −1.35; *p* < 0.00001) and fat mass (SMD −1.52, 95% CI −1.92 to −1.07; *p* < 0.00001). No significant effect on blood glucose, insulin, or lipid profile except triglycerides was found in the analysis. It had no effect on liver and kidney function except that GGT were decreased a little. There were no significant changes in IGF-1 and TNF-α related to tumor growth. Mental health improvement of cancer patients was supported by several trials. Taken together, findings in this study confirmed that the ketogenic diet was a safe approach for cancer patients reducing body weight and fat mass. In addition, cancer treatment-related indicators changed insignificantly. Ketogenic diets may be beneficial to the quality of life of cancer patients. However, intervention duration in most studies is shorter than 6 months, and the effect of a long-term ketogenic diet is still required further validation. More trials with a larger sample size are necessary to give a more conclusive result; PROSPERO registration number: CRD42021277559.

## 1. Introduction

The ketogenic diet is a diet characterized by high fat and low carbohydrate. The traditional ketogenic diets with a ketogenic index of 4:1 consist of 90% fat, 8% protein, and 2% carbohydrates. Various protocols suggested an average of 70–80% fat, 5–10% carbohydrates, and 10–20% proteins, which were found in literature [1]. Under ketogenic diets, the body’s energy source is mainly ketone bodies derived from fat metabolism. This will greatly change the metabolism of the body. In 1921, Wilder RM [2] have found that ketogenic diets could treat refractory epilepsy and reduce the seizure frequency of patients with epilepsy. In recent years, ketogenic diets have had therapeutic effects on multiple neurological disorders, including epilepsy, Alzheimer’s disease, Parkinson’s disease, and glioma [3]. It was also used to try to control the blood glucose level in patients with type 2 diabetes, obesity, hypercholesterolemia, and polycystic ovary syndrome [4,5]. Ketogenic diets are considered to be a safe and effective non-drug treatment for metabolic diseases.

Early in 1930, Warburg found that tumor cells tend to use glycolysis even when oxygen was sufficient. This phenomenon was called a Warburg Effect, which was, aerobic glycolysis [6]. Compared with oxidative phosphorylation, glycolysis produces energy quickly but inefficiently. More than 200-fold glucose is consumed by tumor cells to meet the energy demands of rapid growth compared with normal cells [7]. Moreover, the mitochondria of tumor cells lack the key enzymes to consume ketones, which makes it difficult for tumor cells to obtain energy from ketones. The dependence on aerobic glycolysis makes tumor cells vulnerable to damage in the case of glucose deficiency [8].

Recently, researchers have found that ketogenic diets could effectively improve tumor control and survival time in glioma [9], pancreatic cancer [10], lung cancer [11], prostate cancer [12], and breast cancer [13]. The application of ketogenic diets in tumor treatment has gradually attracted researchers’ attention. Ketogenic diets may change cell metabolism and affect tumor growth. Recent studies have shown that ketogenic diets could not only limit tumor growth, but also increase the tolerance of normal cells to radiotherapy and chemotherapy [14], and enhance the anticancer effects of PD-1 blockade [15].

Cancers bring serious physical and mental burden to patients. Promoting cancer treatment and improving the quality of life of cancer patients are the main goals of cancer researches. Although there have been many studies on ketogenic diets as an adjuvant treatment for cancer, its effect on body composition and metabolic parameters of cancer patients is still unclear. Therefore, this systematic review and meta-analysis was conducted to summarize earlier controlled clinical trials assessing the effect of ketogenic diets on cancer patients’ body composition and metabolic parameters including body weight, body mass index (BMI), fat mass, blood glucose, lipid profiles, and so on.

## 2. Materials and Methods

This systematic review was registered with the International Prospective Register of Systematic Reviews (PROSPERO) on 8 October 2021 (registration number CRD42021277559) and followed the Preferred Reporting Items for Systematic Reviews and Meta-Analyses (PRISMA) guidelines [16].

### 2.1. Data Sources, Search Strategy, and Selection Criteria

We identified relevant studies using medical subject headings (MeSH) and text words related to ketogenic diets and cancer. The systematic literature search was performed in Pubmed, EMBASE, and Cochrane Central Register of Controlled Trials (CENTRAL) until April 2022. Combinations of search terms were shown in Appendix A. There was no restriction of language and time in the selection of literature. To avoid missing any publication, a manual-search was also performed for all reference lists of related clinical trials and reviews to include other potentially eligible trials. Screening and study selection were conducted by two authors (H.Z., H.J.) independently.

### 2.2. Inclusion and Exclusion Criteria

Randomized controlled trials and cohort studies that met the following criteria were selected: participants were adults diagnosed with cancer/tumor; dietary intervention must include ketogenic diets (or the subtype of ketogenic diets). Articles were excluded if they: were non-human species; have no comparison group; were conference abstracts, book chapters, reviews, or other forms without detailed empirical data and have no exposure or outcome of interest.

Based on the above inclusion and exclusion criteria, the titles and abstracts of the selected articles were screened independently by three authors who were not blinded to the authors and the article titles. The full-text versions of potentially eligible articles were retrieved for further evaluation. Any discrepancy that occurred during this process was resolved by consensus.

### 2.3. Data Extraction and Quality Assessment

The two authors (H.Z., H.J.) extracted the relevant data independently using a Microsoft Excel customized sheet for a data extraction based on the PICOS principle. Any discrepancy was settled through joint discussion with the third author (J.X.). The corresponding author was contacted through email for missing data. The following information was extracted: first author, publication year, study design, age of participants, cancer type, details about intervention and control diets, number of participants in both groups, duration of intervention, and outcome (body weight, lipid profile, biochemical indices, etc.). Engauge Digitizer software was used to extract numerical data published as figures in the articles. When the studies measured outcomes in a variety of ways, the result was converted to a uniform scale (mean ± standard deviation).

Risk of bias was assessed according to the Cochrane Handbook recommendations using the “risk of bias” method. Several methodological domains were examined to classify bias: selection bias (random sequence generation and allocation concealment), performance bias (blinding of participants and personnel), detection bias (blinding of outcome assessment), attrition bias (incomplete outcome data), reporting bias (selective reporting), and other bias (carry over effect in cross-over RCTs). Disagreement was resolved through discussion among authors. Additionally, we rated the quality of each evidence as high, moderate, low, and very low in effect estimates for outcomes of change from baseline, based on the Grading of Recommendations, Assessment, Development, and Evaluation (GRADE) approach, in which studies are evaluated on five aspects: risk of bias, consistency of effect, directness, precision, and publication bias [17].

### 2.4. Statistical Analysis

Meta-analyses were conducted using a Review Manager (RevMan) Version 5.4.1 (Revman International, Inc, New York, NY, USA). We used a Cochran–Mantel–Haenszel test and inverse-variance method to perform a meta-analysis. Continuous variables of N, mean, standard deviation (SD), and median (25th percentile, 75th percentile) were extracted from each intervention and control group of the included studies. All the resulting variables were uniformly converted to mean ± SD for merging. For the original study that reported only the median, we converted the median of baseline and post-intervention data to mean ± SD by calculating the closest approximation of mean and SD from the median and interquartile range (IQR) [18,19,20,21]. To do this, standardized mean differences (SMD) and their 95% confidence intervals (CI) were calculated to assess the change in each selected variable. During the analysis process, all the standard errors of the mean (SEM) were converted into SD by using the formula SD = SEM ×$\sqrt{N}$ [22]. Heterogeneity across the studies was quantitatively evaluated by I^2^ index. When I^2^ value > 50%, the random-effects model would be adopted. Otherwise, the fixed effects model would be used. If high heterogeneity was exhibited, the potential source of heterogeneity was explored by analyzing each study. In addition, *p*-value < 0.05 was regarded to be statistically significant.

## 3. Results

### 3.1. Study Selection

A total of 561 articles were identified. After the removal of 66 duplicate records, 495 potential records were left. A total of 340 articles were excluded based on their titles and abstracts. In the left 155 articles, 46 articles are in vitro or animal experimental studies and 31 articles are reviews. There are 20 articles that are not full-text or without public data. In addition, 39 articles were excluded for 9 case reports, 2 protocols, 8 prospective studies, 6 retrospective studies, and 14 non-controlled trials as shown in Figure 1. Two of the rest articles were excluded because the results were irrelevant [23] and intervention did not accord with the inclusive criteria [24]. Flow diagram of the literature search process was illustrated in Figure 1.

Finally, a total of 10 studies and 17 articles were included for meta-analysis. Among the 10 studies, Klement’s three studies were subgroup studies based on different cancer patients in the same study [25].

### 3.2. Study Characteristics

All figures and tables should be cited in the main text as Figure 1, Table 1, etc. The characteristics of eligible studies were summarized in Appendix A. The trials included a total of 495 individuals in which 334 individuals had completed the trials (242 participated in ketogenic diet intervention and 171 completed). The completion rate was 64.43% in the ketogenic diet group and 70.66% in the control group. The rate of each study was shown in Table 1. The inclusion criteria with respect to age, BMI, or cancer type appeared varied in each study, which were summarized in Appendix A.

The comparator group patients received ketogenic diets with more than 55% energy from fat. Intervention in study (Voss 2020 [38,39]) was a ketogenic diet followed with short-term fasting. The study (Kämmerer 2021 [40]) also reported the effect of a low carbohydrates diet intervention, but it was not within the scope of this article. All of the control group patients received standard diets except (Cohen 2018 [26,27,28]), which was an American Cancer Society (ACS) diet. Duration of the trials varied from 1 to 20 weeks. The duration of study (Kämmerer 2021 [40]) was the longest (20 weeks). The effect of 6 and 12 weeks were reported in the study (Khodabakhshi 2019 [31,32,33]). Here, only the data of 12 weeks were extracted and analyzed. The durations of (Augustus 2020) [35], (Kang 2019) [30] and (Cohen 2018 [26,27,28]) were 4, 12, and 16 weeks, respectively. The duration of (Klement 2019 [34], 2020 [36,37], 2021 [41,42]) was 2 to 8 weeks. The duration of (Ok 2018 [29]) and (Voss 2020 [38,39]) was less than 2 weeks.

Among the ten studies, four of them included breast cancer patients, (Khodabakhshi 2019 [31,32,33]), (Klement 2020 [36,37]) and (Kämmerer 2021 [40]). The trial (Cohen 2018 [26,27,28]) included female patients with ovarian or endometrial cancer. The remaining five studies included both male and female patients. The trials (Ok 2018 [29]) and (Kang 2019 [30]) included patients with pancreatic, ampulla of vater, common bile duct and duodenal cancer. Participants in trials (Klement2019 [34], Voss 2020 [38,39], Klement 2021 [41,42]) were head and neck cancer, glioma, and colorectal cancer patients, respectively.

### 3.3. Study Quality of Trials

Individual trial appraisal on each risk of bias was reported in Figure 2. Three of the studies had a high risk of selection bias (Klement 2019 [34], Klement 2020 [36,37], and Klement 2021 [41,42]). The participants were assigned to different groups according to their personal desires. Since the intervention was dietary, the intervention group was informed in all studies (high risk), except (Khodabakhshi 2019 [31,32,33]) (unclear). Placebo was used in control group in this study (Khodabakhshi 2019 [31,32,33]). All of the studies did not mention the blinding of outcome assessment (unclear). The attrition, reporting and other biases were low risk. Otherwise, the risk of bias assessment of the included studies was generally acceptable. The evidence profiles for the comparison ketogenic diets vs. non-ketogenic diets on body weight(moderate), BMI (very low), fat mass (moderate), total cholesterol (low), HDL-C (low), LDL-C (moderate), triglycerides (low), blood glucose (low), insulin (moderate), IGF-1 (moderate), ketone body (low), β-hydroxybutyrate (very low), creatine (very low), and free T3 (very low) are shown in Appendix A.

### 3.4. The Effects of Ketogenic Diets on Body Composition

#### 3.4.1. Body Weight

Eight trials (302 participants) were included in the analysis of body weight outcomes in baseline and post-intervention (Figure 3A). The ketogenic diets group has a small effect size on body weight in post-intervention subgroup (SMD −0.25, 95% CI −0.73 to 0.24, I^2^ = 74%, *p* = 0.32). The overall effects of both subgroups were non-statistic. Both subgroups showed high levels of heterogeneity (baseline: I^2^ = 67%, post-intervention: I^2^ = 74%). The study (Cohen 2018 [26,27,28]) was the main source of heterogeneity. After removing the study (Cohen 2018 [26,27,28]), the I^2^ in baseline and post-intervention were both reduced to 0%, and the effect size in post-intervention was (SMD −0.06, 95% CI −0.31 to 0.19, *p* = 0.61, Appendix A). There was little change from the analysis before.

Four studies reported changes from baseline in body weight. Meta-analysis demonstrated that participants received ketogenic diets were more likely to experience a greater weight reduction compared with those on comparator diets (SMD −1.29, 95% CI −0.73 to 0.24; I^2^ = 79%; *p* < 0.00001, Figure 3B). Heterogeneity originates from (Klement 2019 [34] and Voss 2020 [38,39]) these two studies. There were only five participants in the intervention group of (Klement 2019 [34]), which had a certain impact on the results. The intervention duration of (Voss 2020 [38,39]) was less than 10 days, which might have a weak effect on body weight. After removing the two studies (Klement 2019 [34] and Voss 2020 [38,39]), the effect size changed to (SMD −1.83, 95% CI −2.30 to −1.35, I^2^ = 0%, *p* < 0.00001, Appendix A). It can be inferred from the above results that ketogenic diets had a moderate to large effect on the body weight of cancer patients.

#### 3.4.2. BMI

We found a statistical effect of ketogenic diets on BMI values in post-intervention (SMD −0.51, 95% CI −0.91 to −0.10; 4 trials and 183 participants, I^2^ = 45%, *p* = 0.01, Figure 3C). We analyzed again that BMI in post intervention without study (Voss 2020 [38,39]), and the effect was significant (SMD −0.70, 95% CI −1.04 to −0.36, *p* < 0.0001, I^2^ = 0%, Appendix A). Analysis of BMI change from baseline also revealed a statistical decrease with ketogenic diets interventions (SMD −1.30, 95% CI −1.99 to −0.62, *p* = 0.0002), but only one trial was involved in the analysis (Klement 2021 [41,42]) (Figure 3D).

#### 3.4.3. Fat Mass

Six studies (302 participants) were analyzed for fat mass. The post-intervention value showed ketogenic diets had a small effect on fat mass (SMD −0.25, 95% CI −0.50 to 0.01; I^2^ = 1%; *p* = 0.06, Figure 3E). Analysis of fat mass changes from baseline revealed a beneficial effect of ketogenic diets compared with controls but without significant difference (SMD −0.78, 95% CI −1.63 to 0.07; I^2^ = 82%, *p* = 0.07, Figure 3F). Analysis was repeated after excluding (Klement 2019 [34]); the results appeared to have a large and significant effect (SMD −1.14, 95% CI −1.89 to 0.38; I^2^ = 77%, *p* = 0.01, Appendix A). When the studies (Cohen 2018 [26,27,28] and Klement 2019 [34]) were excluded, the heterogeneity decreased and the overall conclusion showed a statistical difference (SMD −1.52, 95% CI −1.92 to −1.07; I^2^ = 0%, *p* < 0.00001, Appendix A).

### 3.5. Effect on Blood Glucose, Insulin, and IGF-1

#### 3.5.1. Blood Glucose

Seven studies were involved in analysis of blood glucose in baseline and post-intervention. The overall effect in post intervention was small, non-statistic, and high heterogeneity (SMD −0.48, 95% CI −1.04 to 0.09, *p* = 0.10, I^2^ = 82%, Figure 4A). After removing three studies (Augustus 2020 [35] and Khodabakhahi 2019 [31,32,33]), the effect size in post-intervention was little (SMD −0.09, 95% CI −0.47 to 0.29, *p* = 0.64, I^2^ = 46%, Appendix A). Only one study (20 participants) reported changes from the baseline in blood glucose. Analysis of change from baseline of blood glucose showed a statistic and moderate effect (SMD −0.63, 95% CI −1.25 to −0.01, *p* = 0.05, Figure 4B). The number of participants included in the analysis was too small to make definitive conclusions.

#### 3.5.2. Insulin

Seven studies reported insulin changes in the post ketogenic dietary intervention. Analysis of insulin in post-intervention showed a statistic effect (SMD −0.67, 95% CI −1.29 to −0.04, *p* = 0.04, I^2^ = 85%, Figure 4C). The data in study (Kämmerer 2021 [40]) lead to high heterogeneity in analysis of baseline. However, the high heterogeneity did not disappear after removing (Kämmerer 2021 [40]) (Appendix A). Kang 2019 [30] and Klement 2020 [36,37] were also responsible for high heterogeneity in analysis of post intervention. After removing these studies, the high heterogeneity disappeared (SMD −0.75, 95% CI −1.14 to −0.36, *p* = 0.0002, I^2^ = 39%, Appendix A). Meanwhile, the subgroup difference changed to significant (*p* = 0.03). Analysis of change from baseline in insulin included three studies. The effect was little and non-statistic (SMD −0.15, 95% CI −0.49 to 0.19, *p* = 0.38, I^2^ = 0%, Figure 4D). Taken together, we suggested that ketogenic diets might have a small effect on insulin, and the significant difference was generated due to the exclusion of heterogeneous results.

#### 3.5.3. IGF-1

IGF-1 (insulin-like growth factor-1) plays an important role in cell proliferation, differentiation, metabolism, apoptosis, and angiogenesis. Six studies (283 patients) were included in the analysis of post-intervention in IGF-1(SMD −0.13, 95% CI −0.47 to 0.21, *p* = 0.45, I^2^ = 52%, Figure 4E). Removing the study (Kämmerer 2021 [40]), which was the main reason for the heterogeneity of analysis in baseline, the effect of ketogenic diets in post-intervention was small and statistical (SMD −0.27, 95% CI −0.53 to −0.01, *p* = 0.04, I^2^ = 0%, Appendix A). Three studies reported the change from baseline in IGF-1. The effect was (SMD −0.03, 95% CI −0.62 to 0.55, *p* = 0.91, I^2^ = 66%, Figure 4F). According to the data in the article (Voss 2020 [38,39]), we found that IGF-1 decreased in ketogenic diets intervention group but increased in the non-ketogenic diets group. This is inconsistent with the data of change from baseline. After removing this study (Voss 2020 [38,39]), analysis of change from baseline showed that ketogenic diets had a small effect on IGF-1, but without a significant difference (SMD −0.30, 95% CI −0.70 to 0.10, *p* = 0.14, I^2^ = 0%, Appendix A).

### 3.6. Effects of Ketogenic Diets on Lipid Profiles

#### 3.6.1. Total Cholesterol

Based on data from seven trials (261 participants), we found a moderate effect of ketogenic diets on total cholesterol in post-intervention values (SMD 0.69, 95% CI −0.19 to 1.56; *p* = 0.12, I^2^ = 90%, Figure 5A). However, subgroup differences were not significant comparing effects of baseline and post intervention (*p* = 0.68). Differences in baseline data among study participants may be the reason for the high heterogeneity. After removing the study (Augustus 2020 [35]) and (Kämmerer 2021 [40]), the I^2^ in baseline analysis reduced to 37%. The heterogeneity in post intervention also disappeared, and the differences between subgroups were still not significant (*p* = 0.86, Appendix A). Two studies reported the change from baseline in total cholesterol. Analysis of change from baseline showed that ketogenic diets intervention had no effect on total cholesterol (SMD 0.10, 95% CI −0.33 to 0.52, *p* = 0.66, I^2^ = 0%, Figure 5B).

#### 3.6.2. HDL-C

Analysis results of HDL-C were illustrated in Figure 4C,D. The effect size of post intervention was moderate, non-statistic, and had high heterogeneity (SMD 0.48, 95% CI −0.32 to 1.28, *p* = 0.24, I^2^ = 87%, Figure 5C). Data in the study (Kämmerer 2021 [40]) caused high heterogeneity in the analysis of baseline. After removing the study (Kämmerer 2021 [40]), the analysis result in baseline and post intervention were (SMD 0.07, 95% CI −0.28 to 0.43, *p* = 0.69, I^2^ = 26%) and (SMD 0.14, 95% CI −0.17 to 0.45, *p* = 0.38, I^2^ = 0%), respectively (Appendix A). The effect size of change from baseline was small (SMD 0.29, 95% CI −0.26 to 0.65, *p* = 0.30, I^2^ = 58%, Figure 5D). After removing the study (Klement 2021 [41,42]), I^2^ decreased to 0%, and SMD was 0.02 (95% CI −0.41 to 0.44, *p* = 0.94, Appendix A). It can be inferred that ketogenic diets had little effect on HDL in cancer patients.

#### 3.6.3. LDL-C

Six studies (220 participates) were included in the analysis in post intervention on LDL-C. Although the analysis results of post intervention in LDL-C suggested a large effect (SMD 0.93, 95% CI −0.01 to 1.86, *p* = 0.05, I^2^ = 90%, Figure 5E), there was no statistic subgroup difference between baseline and post intervention (*p* = 0.39). After removing the study (Kämmerer 2021 [40]), I^2^ in post intervention decreased to 0%, and the effect size became small (SMD 0.35, 95% CI 0.05 to 0.65, *p* = 0.65, Appendix A). The subgroup difference was non-statistic (*p* = 0.72). Analysis of change from baseline supports the point above (SMD 0.03, 95% CI −0.46 to 0.52, *p* = 0.91, I^2^ = 48%, Figure 5F). Therefore, we reasoned that the ketogenic diets had little effect on LDL-C.

#### 3.6.4. Triglycerides

Analysis of triglycerides in post intervention appeared to a large effect (SMD −0.81, 95% CI −1.71 to 0.08, *p* = 0.07, I^2^ = 89%, Figure 5G). However, there was no statistic subgroup difference (*p* = 0.81). The heterogeneity of baseline was very high, which might be due to the heterogeneity of participants’ data itself (Kämmerer 2021 [40]). After removing the study (Kämmerer 2021 [40]) and (Klement 2021 [41,42]), the effect size in post intervention changed to small (SMD −0.20, 95% CI −0.67 to 0.28, *p* = 0.41, I^2^ = 44%, Appendix A). Analysis of change from baseline showed a small effect (SMD −0.37, 95% CI −0.89 to 0.15, *p* = 0.03, I^2^ = 67%). After removing the study (Cohen 2018 [26,27,28]), the effect of ketogenic diets on triglycerides was moderate and statistically significant (SMD −0.59, 95% CI −1.00 to −0.18, *p* = 0.005, I^2^ = 30%, Appendix A).

### 3.7. Ketosis

#### 3.7.1. Ketone Body and Ketosis Events

Ketone bodies are a product of fatty acid metabolism, which is composed of acetoacetic acid, β- hydroxybutyrate, and acetone. The most commonly used test method of ketone bodies is a urine test. The detection of blood ketone bodies is more accurate. Six studies reported detection of ketone body or β-hydroxybutyrate. Based on the subgroup analysis of baseline and post intervention, Ketogenic diets seemed to have a non-statistical effect on ketone body (SMD 0.71, 95% CI −1.53 to 2.95, *p* = 0.53, Figure 6A). However, in the study (Kang 2019 [30]), the concentration of blood ketone body in participants was high before intervention, which confused the analysis results. Meanwhile, the blood ketone body concentration of participants in ketogenic diets group increased significantly in the second week, but decreased in the fourth week (Kang 2019 [30]). After removing the study, we found that ketogenic diets had a statistic effect on ketone body (SMD 1.83, 95% CI 1.22 to 2.44, *p* < 0.00001, Appendix A). Analysis of change from baseline confirmed this (SMD 1.25, 95% CI 0.27 to 2.23, *p* = 0.01, Figure 6B). Ketosis events were reported in two studies (Cohen2018 [26,27,28] and Ok 2018 [29]). Analysis of Ketosis events showed that ketogenic diets increased the risk of ketosis (Odds rate 7.11, 95% CI 1.97 to 25.72, *p* = 0.003, Figure 6C).

#### 3.7.2. β-Hydroxybutyrate

β-hydroxybutyrate was one of the ketone body components, signal of ketosis. Five studies (182 participants) were included in analysis of β-hydroxybutyrate. Although the subgroup of post intervention showed a large effect (SMD 0.85, 95% CI 0.30 to 1.41, *p* = 0.002, I^2^ = 63%, Figure 6B), the analysis of baseline also showed a small effect (SMD 0.48, 95% CI 0.18 to 0.78, *p* = 0.002, Figure 6B). The subgroup difference was not significant (*p* = 0.24). After removing the study (Klement 2021 [41,42]), the analysis result of post intervention was consistent with that before removal (SMD 0.65, 95% CI 0.20 to 1.09, *p* = 0.004, I^2^ = 29%, Appendix A). In addition, the subgroup difference remained non-significant (*p* = 0.55, Appendix A). The analysis results of change from the baseline suggested a statistic and moderate effect (SMD 0.79, 95% CI 0.26 to 1.32, *p* = 0.004). However, only one study included in the analysis.

### 3.8. Renal Function Test

#### 3.8.1. Creatinine

Analysis of creatinine included five studies (223 participants). The analysis result of post intervention showed a moderate effect (SMD −0.52, 95% CI −1.19 to 0.15, *p* = 0.13, I^2^ = 82%, Figure 7A). Heterogeneity in both the baseline and post intervention was very high. In order to remove a potential confounding factor, we repeated meta-analysis after excluding the study (Khodabakhshi 2019 [31,32,33]), which caused the high heterogeneity in analysis of the baseline. I^2^ in analysis of baseline reduced to 33%, while in post intervention was 86% (SMD −0.45, 95% CI −1.37 to 0.47, *p* = 0.34, Appendix A). In Kang’s study, the detection method of creatinine was LC-MS, which was different from other studies. Creatinine concentration decreased in the second week of ketogenic diets intervention, but increased in the fourth week. After removing the study (Khodabakhshi 2020 [31,32,33]) and (Kang 2019 [30]), the analysis result of post intervention showed a large and statistic significant effect (SMD −0.86, 95% CI −1.65 to −0.06, *p* = 0.04, I^2^ = 80%, Appendix A). High heterogeneity was still existing. Only one study reported the change from baseline in creatinine. Analysis of it showed a small but non-statistical effect (SMD −0.46, −0.98 to 0.04, *p* = 0.09, Figure 7B).

#### 3.8.2. BUN and Urea and Uric Acid

Figure 7C illustrated the analysis results of blood urea nitrogen (BUN) in baseline and post intervention. Four studies and 205 participants were included in the analysis. Although the analysis of post intervention showed a small and statistic significant effect (SMD 0.34, 95% CI 0.07 to 0.62, *p* = 0.02), there was no significant difference between subgroup (*p* = 0.29). Analysis of urea in baseline and post intervention was illustrated in Figure 7D. The analysis results suggested ketogenic diets had no effect on urea (SMD −0.13; 95% CI −0.47 to 0.20, *p* = 0.43). Two studies and 106 participants were included in the analysis of uric acid in Figure 7E. A small to moderate effect with high heterogeneity was found in analysis of post intervention (SMD 0.50, 95% CI 0.05 to 0.96, *p* = 0.03, I^2^ = 94%). By analyzing the two studies separately, we found no evidence of an effect of ketogenic diets on uric acid. There were no significant differences between subgroups (*p* = 0.27).

### 3.9. Liver Function Test

Analysis results of alanine aminotransferase (ALT), aspartate transaminase (AST), albumin, and gamma-glutamyl transpeptidase (GGT) were illustrated in Figure 8. No effect of ketogenic diets was found on ALT (SMD 0.03, 95% CI −0.25 to 0.3, *p* = 0.55, I^2^ = 0%), AST (SMD −0.09, 95% CI −0.40 to 0.22, *p* = 0.95), and albumin (SMD 0.24, 95% CI −0.04 to 0.51, *p* = 0.76). The effect size of analysis in the GGT baseline was moderate (SMD 0.69, 95% CI 0.20 to 1.17, *p* = 0.13) and in post intervention was small (SMD −0.22, 95% CI −0.54 to 0.11, *p* = 0.69). The subgroup difference was statistical (*p* = 0.002). GGT might be decreased by ketogenic diets intervention.

### 3.10. Free T3 and TNF-α

Two studies were included in analysis of free T3 and TNF-α (Figure 9). The effect size in baseline and post intervention was (SMD −0.07, 95% CI −0.47 to 0.32) and (SMD −0.61, 95% CI −1.01 to −0.20), respectively. Comparing the analysis results of baseline and post, we inferred that ketogenic diets might decrease free T3 (*p* = 0.06). The analysis of change from the baseline in free T3 confirmed this point (SMD −0.75, 95% CI −1.42 to −0.09, *p* = 0.03). No effect of ketogenic diets was found on TNF-α (SMD 0.02, 95% CI −0.42 to 0.47, *p* = 0.44).

### 3.11. Quality of Life

The effects of ketogenic diets on the quality of life of cancer patients are summarized in Table 2. Two trials (Kang 2019 [30] and Klement 2019 [34]) did not report results on quality of life. Two trials (Ok 2018 [29] and Voss 2020 [38,39]) indicated no significant difference in outcomes. The remaining six trials put forward positive conclusions that ketogenic diets are beneficial to quality of life of cancer patients. These benefits included enhancing mental health and physical health, reducing side effects, and so on. Five of the six studies (except Cohen 2018 [26,27,28]) considered that ketogenic diets could improve mental health. Confusingly, in the study (Khodabakhshi 2019 [31,32,33]), ketogenic diets improved the quality of life compared with the control diet at 6 weeks, but there was no significant difference between groups at 12 weeks.

## 4. Discussion

As is known to all, we are what we eat. Diet plays an important role in health. Many studies have shown that a high-fat diet can become an inducement and promotion of cancer [43,44,45,46]. What about ketogenic diets? In this meta-analysis of 10 controlled clinical trials, we investigated the effect of ketogenic diets relative to general diets on body composition and characteristics in cancer patients. Body weight, BMI, and fat mass were reduced post ketogenic diets intervention. Blood glucose and insulin were decreased but without a statistical difference. Little effect was found on a blood lipid profile. The blood ketone bodies level was raised significantly, and ketosis occurred. Ketogenic diets had little effect on liver and kidney function. GGT was decreased in a certain degree. A significant decrease in free T3 was also found after ketogenic diets intervention. In addition, more trials agreed that ketogenic diets were beneficial to the quality of life of cancer patients.

In recent years, the association between obesity and cancer incidence rate and mortality has been well confirmed [47,48]. From a prospective cohort study of more than 900,000 American adults, it could be speculated that obesity increased the risk of death from cancer [49]. Many subsequent studies confirmed these findings. In 2012, cancers attributed to overweight accounted for about 3.9% (544,300) of all cancer cases worldwide. The statistical results suggest that there is a causal relationship between obesity and the risk of at least 13 cancers [50]. Research on adjusting diet to promote cancer treatment is growing vigorously. Reducing carbohydrate intake and improving fat supply will force the body to metabolize fat for energy. Ketogenic diets are such a process that simulates the metabolic state of fasting. Several clinical trials have shown that ketogenic diets could reduce obesity representation of participants. Body weight, body mass index, fat mass, and waist circumference could be significantly reduced [51,52,53,54,55]. Patients with NAFLD undergoing a ketogenic diet achieved superior weight loss, with significant visceral adipose tissue and liver fat fraction reductions when compared to the standard diet [56]. Ketogenic diets might be an alternative dietary approach to decrease fat mass and visceral adipose tissue without decreasing lean body mass [57,58]. Our meta-analysis confirmed that ketogenic diets could decrease the fat mass of cancer patients.

In recent years, control of blood glucose has become the focus of ketogenic diet researches. In a three-month ketogenic diet study involving 55 cancer patients, it was found that the total ketone body increased significantly, and the levels of fasting blood glucose and insulin decreased significantly. No serious adverse events related to diet were observed [59]. A one month single-arm prospective study demonstrated that a ketogenic diet was generally safe for patients with high grade gliomas. It changed metabolism, increased blood ketone body, and decreased level of insulin. No change in fasting blood glucose was observed [60]. A meta-analysis of the effect of ketogenic diets on blood glucose in patients with obesity or overweight showed that there was no significant decrease in blood glucose index in a ketogenic diet group compared with the control group. Consistent with our results, ketogenic diets had little effect on decreasing the blood glucose of cancer patients.

Will a high-fat ketogenic diet affect liver and kidney function? A prospective observational real-life study was conducted on patients with obesity and mild kidney failure undergoing a 3-month ketogenic diet. The study found that weight and fat mass were significantly reduced, and the metabolic parameters and metabolic rate were significantly improved. There were no clinically relevant changes in liver and kidney function, and no differences were found in efficacy and safety results. More than a quarter of patients with mild renal failure reported normalization of glomerular filtration after dietary intervention [61]. Our meta-analysis also found that ketogenic diets had no effect on liver and kidney function, except the GGT level.

IGF-1 has been proved to increase the proliferation rate of many cancers and lead to treatment resistance [62,63]. IGF-1 levels in patients with Laron syndrome are very low and protected them from cancer [64], as is the case in animal models with low IGF-1 [65]. However, a significant effect of ketogenic diets on IGF-1 was not found in this meta-analysis and neither was TNF-α.

Clinical trials have also proved that ketogenic diets could improve the quality of life of patients with breast cancer [31,36]. Findings of a retrospective single-center study suggested that metabolically supported chemotherapy with ketogenic diets may bring about improvements in survival outcomes and treatment response rates in metastatic NSCLC and pancreatic cancer, without additional safety concerns [11,66]. Other studies have shown that ketogenic diet as an adjuvant to standardized chemo-radiation treatment for glioblastoma multiforme was safe and feasible, but it had no substantial changes in patients’ quality of life, neurological function, and impairment [67]. In this study, we find that most studies considered that ketogenic diets had benefits on quality of life of cancer patients. Improvements in mental health were well recognized. Although there have been many studies on ketogenic diets as an adjuvant treatment for cancer, more solid evidence is still needed [1].

Compared with previous similar meta-analysis, our analysis included more newly studies and statistics. In order to make the results more accurate and reliable, we excluded the study of low-carbon diet and included the data of ketogenic diets intervention group only. However, there are still several limitations in this meta-analysis. The first is that a blinded outcome assessment could not be performed because the intervention in trials was dietary modification. Meanwhile, diet was not strictly regulated and ingredients were not uniform. These were detrimental to the accuracy of the trials results. Secondly, there was heterogeneity in the results of the analysis. In addition to patients’ physical differences, patients with different cancers and stages will also lead to heterogeneity. Subgroup analysis in different cancers, cancer stages, and treatments are still lacking. Thirdly, the available RCTs are still small enough to support us from making a firm conclusion. It needs more RCTs on ketogenic diets to support. In addition, considering that duration of intervention in most studies was less than 6 months, further research is needed to examine the maintenance of the effect in a long period of time.

## 5. Conclusions

Based on the results of current meta-analysis, we speculated that the ketogenic diet is safe for cancer patients to reduce their body weight and fat mass. Dietary compliance was lower in the ketogenic diets group than in the general diets. Ketogenic diets had no significant effect on blood glucose, insulin, and lipid profile except triglycerides. GGT decrease was found in the analysis of liver and kidney function changes. There were no significant changes in IGF-1 and TNF-α related to tumor growth. Ketogenic diets may be beneficial to the quality of life of cancer patients. There was inadequate evidence to support the beneficial effects of ketogenic diets on cancer treatment. Large comparative studies are warranted to draw robust conclusions.

## Figures and Tables

**Figure 1 nutrients-14-04192-f001:**
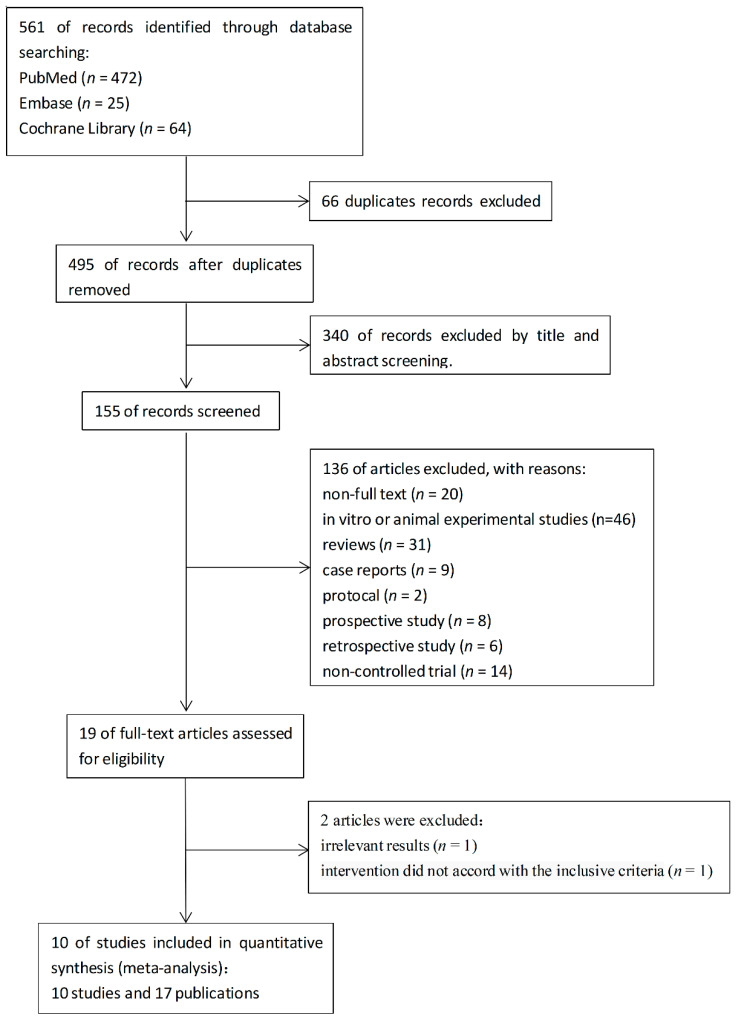
Flow diagram of the literature search process.

**Figure 2 nutrients-14-04192-f002:**
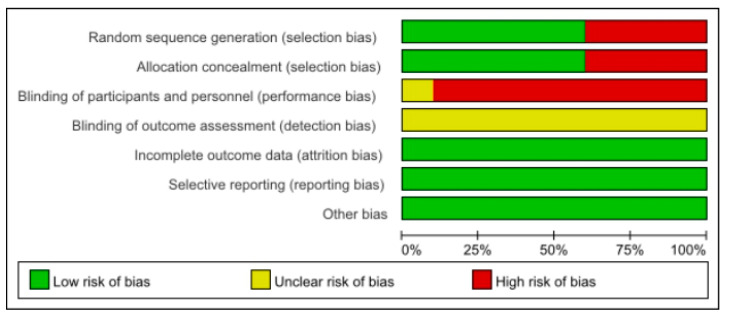
Risk of bias assessment of trials.

**Figure 3 nutrients-14-04192-f003:**
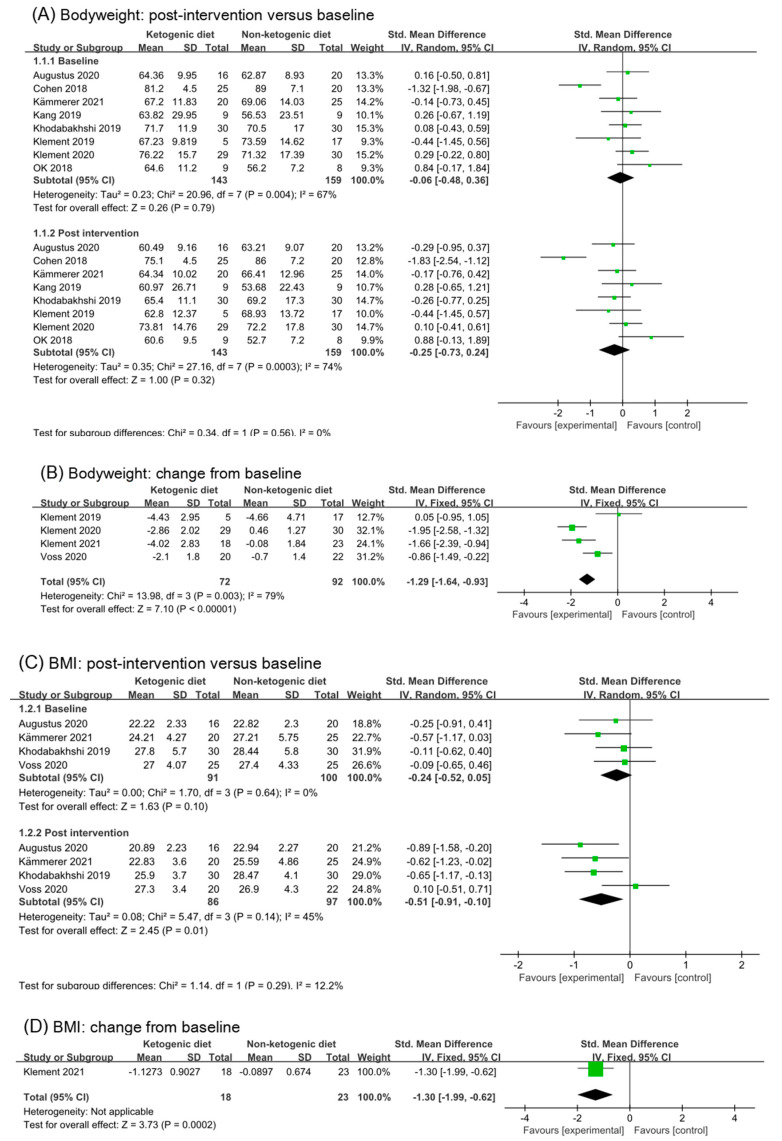
Forest plot for the effects of ketogenic diets on body weight, BMI and fat mass. (**A**,**B**) were subgroup effects of the baseline versus post-intervention and effect of change from baseline on body weight; (**C**,**D**) were ones on BMI; (**E**,**F**) were ones on fat mass [26,27,28,29,30,31,32,33,34,35,36,37,38,39,40,41,42].

**Figure 4 nutrients-14-04192-f004:**
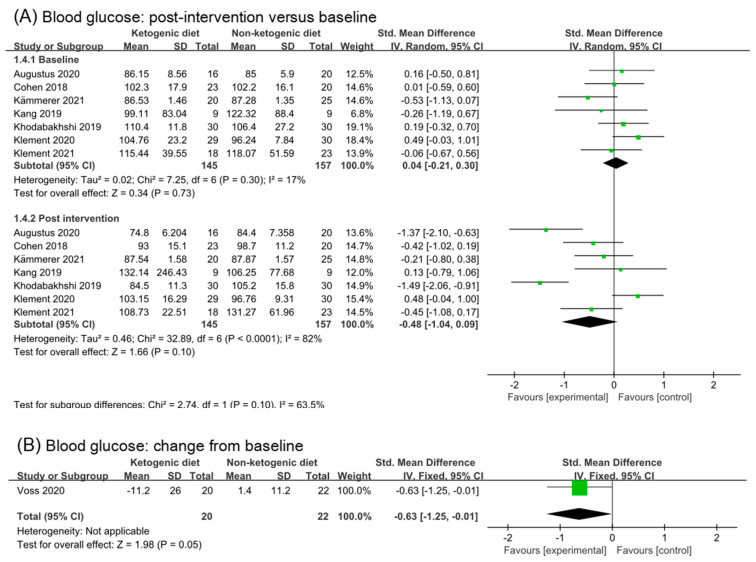
Forest plot for the effects of ketogenic diets on blood glucose, insulin and IGF-1. (**A**,**B**) were subgroup effects of the baseline versus post-intervention and effect of change from baseline on blood glucose; (**C**,**D**) were ones on insulin; (**E**,**F**) were ones on IGF-1 [26,27,28,30,31,32,33,35,36,37,38,39,40,41,42].

**Figure 5 nutrients-14-04192-f005:**
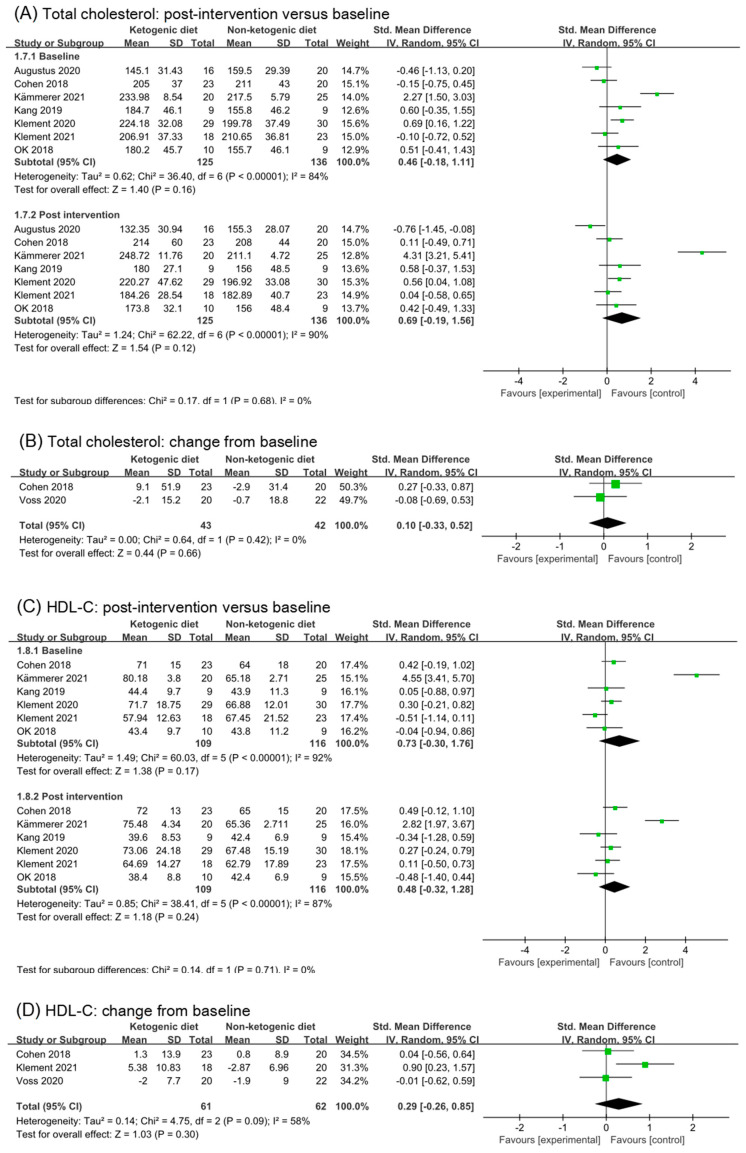
Forest plot for the effects of ketogenic diets on lipid profiles. (**A**,**B**) were subgroup effects of the baseline versus post-intervention and effect of change from baseline on total cholesterol; (**C**,**D**) were ones on HDL-C; (**E**,**F**) were ones on LDL-C; (**G**,**H**) were ones on triglycerides [26,27,28,29,30,31,32,33,35,36,37,38,39,40,41,42].

**Figure 6 nutrients-14-04192-f006:**
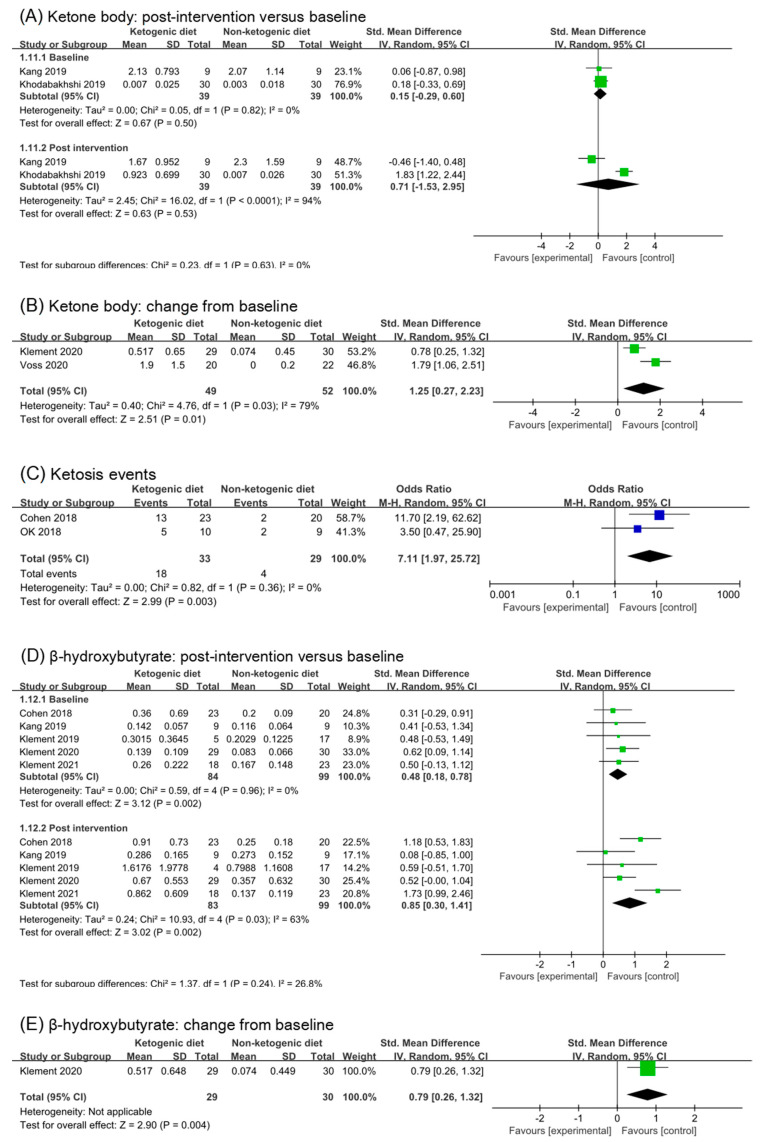
Forest plot for the effects of ketogenic diets on ketone bodies, ketosis and β-hydroxybutyrate. (**A**,**B**) were subgroup effects of the baseline versus post-intervention and effects of change from baseline on ketone body; (**D**,**E**) were ones on β-hydroxybutyrate; (**C**) was risk ratio for ketosis in comparison of ketogenic group and control group [26,27,28,29,30,31,32,33,34,35,36,37,38,39,41,42].

**Figure 7 nutrients-14-04192-f007:**
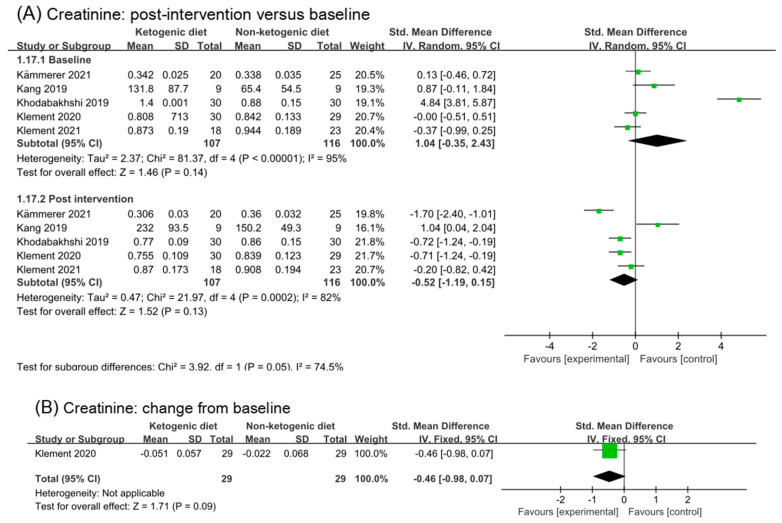
Forest plot for the effects of ketogenic diets on renal function. (**A**,**C**–**E**) were subgroup effects of the baseline versus post-intervention on creatinine, BUN, urea and uric acid; (**B**) was the effect of change from baseline on creatinine [31,32,33,36,37,38,39,40,41,42].

**Figure 8 nutrients-14-04192-f008:**
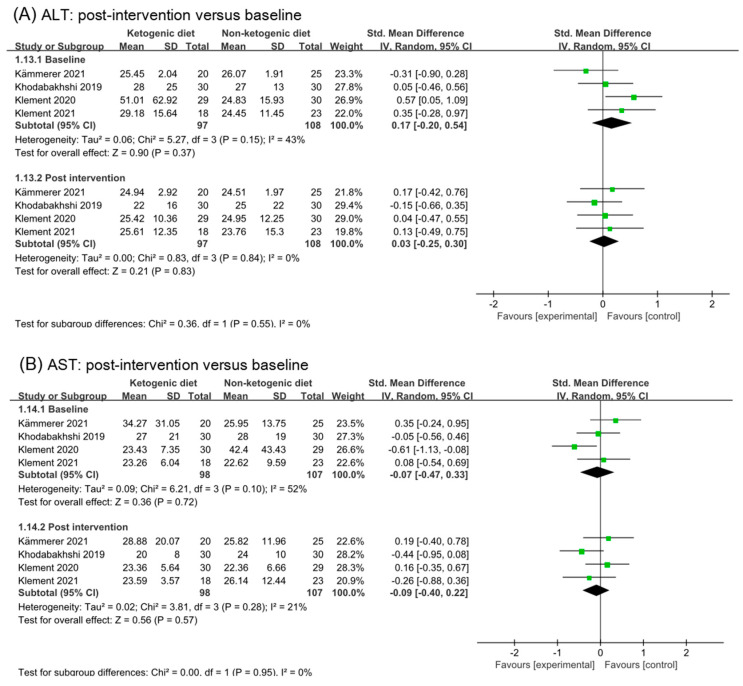
Forest plot for the effects of ketogenic diets on liver function. (**A**–**D**) were subgroup effects of the baseline versus post-intervention on ALT, AST, albumin, and GGT [31,32,33,36,37,40,41,42].

**Figure 9 nutrients-14-04192-f009:**
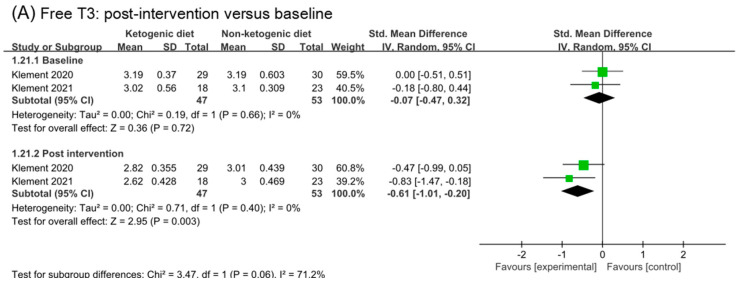
Forest plot for the effects of ketogenic diets on free T3 and TNF-α. (**A**,**C**) were subgroup effects of the baseline versus post-intervention on free T3 and TNF-α; (**B**) was the effect of change from baseline on free T3 [30,31,32,33,36,37,41,42].

**Table 1 nutrients-14-04192-t001:** Completion rate of studies.

Author, Year	Diet Type	Number of Enrollment	Number of Completion	Completion Rate	Special Reasons for Withdrawal
Cohen 2018 [26,27,28]	ACS	36	20	55.56%	4
	KD	37	25	67.57%	5
Ok 2018 [29]	GD	10	9	90.00%	0
	KD	20	10	50.00%	4
Kang 2019 [30]	GD	10	9	90.00%	1
	LCKD	20	9	45.00%	6
Khodabakhshi 2019 [31,32,33]	SD	37	30	81.08%	2
	KD	40	30	75.00%	0
Klement 2019 [34]	SD	17	17	100.00%	0
	KD	5	5	100.00%	0
Augustus 2020 [35]	SD	20	20	100.00%	0
	MKD	20	16	80.00%	1
Klement 2020 [36,37]	SD	31	30	96.77%	0
	KD	32	29	90.63%	0
Voss 2020 [38,39]	SD	25	22	88.00%	0
	KD-IF	25	20	80.00%	2
Kämmerer 2021 [40]	SD	31	26	83.87%	5
	KD	30	24	80.00%	3
Klement 2021 [41,42]	SD	25	23	92.00%	1
	KD	24	19	79.17%	3

Abbreviations: American Cancer Society (ACS) diet, ketogenic diet (KD), general diet (GD), low-carbohydrate ketogenic diet (LCKD), modified ketogenic diet (MKD), standard diet (SD).

**Table 2 nutrients-14-04192-t002:** Summary of quality of life in studies.

Author, Year	KD versus Non-KD
Cohen 2018 [26,27,28]	The physical component summary scores at 12 weeks were significantly higher in the KD than that of the ACS. No difference in mental component summary.
Ok 2018 [29]	No significant difference.
Kang 2019 [30]	Not reported.
Khodabakhshi 2019 [31,32,33]	A higher global quality of life and physical activity scores compared to the control group at 6 weeks. No significant difference at 12 weeks.
Klement 2019 [34]	Not reported.
Augustus 2020 [35]	Patients in KD group had an improvement over time in their self-reported quality of life as well as mental health.
Klement 2020 [36,37]	In the KD group, emotional functioning was improved, and insomnia and systemic therapy side effects decreased significantly during the study.
Voss 2020 [38,39]	No significant difference.
Kämmerer 2021 [40]	Compared with SD group, KD group was able to improve their emotional and social functioning scores as well as reduce their bloating symptom score during the study. The decline of several functioning scores that occurred in both groups was consistently less severe in the KD group. In particular, physical and role functioning had decreased significantly only in the SD group.
Klement 2021 [41,42]	KD improved overall quality of life significantly and remained the highest. In addition, the KD group also achieved improvements in emotional functioning and insomnia.

## Data Availability

Data described in the manuscript will be made publicly and freely available without restriction.

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
