# Peer review of "Effect of Ketogenic Diets on Body Composition and Metabolic Parameters of Cancer Patients: A Systematic Review and Meta-Analysis"

_nutrients, 2022, doi:10.3390/nu14194192_

Round 1
Reviewer 1 Report
This is a generally reasonable meta-analysis of the available data on the use of a ketogenic diet (KD) in patients with cancers. There are a few things that should be addressed:
1. Reference 4 is out of date.
2. There are a number of places where the authors provide contradictory information without adequate discussion. For example, Intro lines 46-50, discussion lines 478-480 vs lines 486-7 regarding glucose reduction. Could this be due to the specific diets employed, the duration of the diet, etc. Were there differences in the results reported regarding patient survival, quality of life, etc. If these data are not available that should be stated. Since this is one of the reasons the KD has been employed, this is an important point. The authors provide some comments spread in the discussion, but this should be strengthened.
3. The data from Kang et al was problematic in a number of the analyses. Was there something in the work that would explain the aberrant results?
4. It would be easier to read the forest plots if the titles of the plots (ie total cholesterol, glucose, etc) were above each one instead of only in the figure legends.
5. Reference 67 is not in the manuscript and is probably an error.
6. There are a number of typos (ie line 14 “low carbon diet”. The presence or absence of a space between text and reference numbers is inconsistent.
Author Response
Responses to the comments from Reviewer 1:
This is a generally reasonable meta-analysis of the available data on the use of a ketogenic diet (KD) in patients with cancers. There are a few things that should be addressed:
- Reference 4 is out of date.
Answer: Thanks for your comment. We have replaced it with a literature published in 2019.
- There are a number of places where the authors provide contradictory information without adequate discussion. For example, Intro lines 46-50, discussion lines 478-480 vs lines 486-7 regarding glucose reduction. Could this be due to the specific diets employed, the duration of the diet, etc. Were there differences in the results reported regarding patient survival, quality of life, etc. If these data are not available that should be stated. Since this is one of the reasons the KD has been employed, this is an important point. The authors provide some comments spread in the discussion, but this should be strengthened.
Answer: Thanks for your comment. The more accepted theory, based on studies in patients with diabetes or obesity, is that the ketogenic diets can control blood glucose level without rising. However, the effect of lowering blood glucose may vary due to specific diets, living habits and individual differences. Our analysis results show that the effect of ketogenic diets on blood glucose of cancer patients is little. And it is undeniable that a small number of participants are complicated with diabetes or obesity, which will have a certain impact on the results. A further increase in the number of included studies is needed to provide adequate evidence. We have also revised the contradictory words.
Since the relationship between ketogenic diet and survival rate was not discussed in the literature, we have no way to show it. We have added a section 3.11. Quality of life in manuscript. Most trials considered that ketogenic diets might be beneficial to quality of life.
- The data from Kang et al was problematic in a number of the analyses. Was there something in the work that would explain the aberrant results?
Answer: Thanks for your comment. We believe that there are few participants in this study, and a single variable may have a great impact on the overall mean value. In addition, the detection method of metabolites was LC-MS, which was different from other studies.
- It would be easier to read the forest plots if the titles of the plots (ie total cholesterol, glucose, etc) were above each one instead of only in the figure legends.
Answer: Thanks for your suggestion. We have added them into the figures.
- Reference 67 is not in the manuscript and is probably an error.
Answer: Thanks for your comment. We have checked and added the reference.
- There are a number of typos (ie line 14 “low carbon diet”. The presence or absence of a space between text and reference numbers is inconsistent.
Answer: Thanks for your comment. We have checked and revised the manuscript.
Reviewer 2 Report
According to the authors, in this meta-analysis study, it was evaluated the effects of ketogenic diets on cancer patients. Despite the number of studies used to perform the present study has been small, interesting data were presented, particularly and, as expected, in terms of anthropometric parameters and ketosis. However, there are some points that should be concerned, too.
In the Abstract section:
1) The authors need to clarify why applying a ketogenic diet to cancer patients is important.
In the Introduction section:
2) Although the authors presented pieces of information that can support the purpose that the alteration of diet can impact tumor cells, the comment regarding the use of the ketogenic diet, shown in the last sentence (lines 47-50) in the first paragraph on page 2, needs of one reference to support this idea.
3) In this section the authors highlighted the potential effects of the ketogenic diet on the metabolism of tumor and non-tumor cells. However, the authors did not show sufficient information, regarding the relevance of this diet in body composition and some systemic metabolites (glucose and lipid profile), as well as inflammatory status, that allowed us to understand the real importance of the present study. So, It is recommended to add some pieces of information clarifying the relevance of the application of this diet in the parameters assessed in this study.
4) How many cancer patients present overweight and obese? And also, do all of these individuals with overweight or obese, and who have cancer, could be benefit from the ketogenic diet?
In the Results section
5) In Figure 1, why after the exclusion of 135 articles the authors described that 19 articles, and not 20, were eligible?
6) Please, correct the word “slection” to “selection” in line 188.
7) Please, add the p-values in the results presented on lines 295, 300, 325, 326, 328. I recommend carefully checking the presentation of p-values in all parameters assessed.
In the Discussion section:
8) Although the authors presented some interesting data, the main outcomes found were expected, since the main effects of the ketogenic diet are reductions in body weight, fat mass, and BMI, in association or not with the increase in ketonic bodies. However, in these analyses, it was found more significant results when one or more studies were removed from the analysis. So, it is clear that even though the effect of the ketogenic diet was reinforced here, the impact of this diet on cancer patients is unclear.
Therefore, it is also paramount to discuss whether this heterogenicity of the results could be associated with the type of cancer in which the study was performed.
9) Another point that needs to be more explained is related to the real effect of the use of the ketogenic diet in cancer patients, since, as presented, the main results are associated with anthropometric alteration and not metabolic parameters. In addition, what is the percentage of overweight and obesity in cancer patients? Maybe these data can help the authors to sustain the relevance of the study.
Author Response
Responses to the comments from Reviewer 2:
According to the authors, in this meta-analysis study, it was evaluated the effects of ketogenic diets on cancer patients. Despite the number of studies used to perform the present study has been small, interesting data were presented, particularly and, as expected, in terms of anthropometric parameters and ketosis. However, there are some points that should be concerned, too.
In the Abstract section:
1) The authors need to clarify why applying a ketogenic diet to cancer patients is important.
Answer: Thanks for your suggestion. “A ketogenic diet characterized by high fat and low carbohydrate can drive the body to produce a large number of ketone bodies, altering human metabolism. Unlike normal cells, tumor cells have difficulty in consuming ketone bodies. Therefore, the application of ketogenic diets in cancer therapy is gaining attention. However, the effect of ketogenic diets on body parameters of cancer patients is not well established. This meta-analysis aimed to summarize the effects of ketogenic diets on cancer patients in earlier controlled trials.” We have added these words to the abstract.
In the Introduction section:
2) Although the authors presented pieces of information that can support the purpose that the alteration of diet can impact tumor cells, the comment regarding the use of the ketogenic diet, shown in the last sentence (lines 47-50) in the first paragraph on page 2, needs of one reference to support this idea.
Answer: Thanks for your comment. We have added the reference.
3) In this section the authors highlighted the potential effects of the ketogenic diet on the metabolism of tumor and non-tumor cells. However, the authors did not show sufficient information, regarding the relevance of this diet in body composition and some systemic metabolites (glucose and lipid profile), as well as inflammatory status, that allowed us to understand the real importance of the present study. So, It is recommended to add some pieces of information clarifying the relevance of the application of this diet in the parameters assessed in this study.
Answer: Thanks for your suggestion. The aim of this meta-analysis was to explore the effects of ketogenic diets on body parameters of cancer patients, providing evidence that ketogenic diets can improve physical health of cancer patients. We have added more information to clarify this.
4) How many cancer patients present overweight and obese? And also, do all of these individuals with overweight or obese, and who have cancer, could be benefit from the ketogenic diet?
Answer: Thanks for your comment. We are sorry that we don’t have data on individual patients, so we cannot give the number of obese patients. Since the lowest baseline BMI was 26.8 in the trials included in the analysis of fat mass change, we considered the majority of patients were overweight, and some were obese (BMI >30). We have no evidence that all obese patients benefit from diets, but we can be sure that most of them do.
In the Results section
5) In Figure 1, why after the exclusion of 135 articles the authors described that 19 articles, and not 20, were eligible?
Answer: Thanks for your comment. The number was typed incorrectly. We have corrected it to 136. Thanks again.
6) Please, correct the word “slection” to “selection” in line 188.
Answer: Thanks for your comment. We have corrected it.
7) Please, add the p-values in the results presented on lines 295, 300, 325, 326, 328. I recommend carefully checking the presentation of p-values in all parameters assessed.
Answer: Thanks for your suggestion. We have checked the whole manuscript and added the missing p-values.
In the Discussion section:
8) Although the authors presented some interesting data, the main outcomes found were expected, since the main effects of the ketogenic diet are reductions in body weight, fat mass, and BMI, in association or not with the increase in ketone bodies. However, in these analyses, it was found more significant results when one or more studies were removed from the analysis. So, it is clear that even though the effect of the ketogenic diet was reinforced here, the impact of this diet on cancer patients is unclear.
Therefore, it is also paramount to discuss whether this heterogeneity of the results could be associated with the type of cancer in which the study was performed.
Answer: Thanks for your comment. Yes, we strongly agree with you. Different cancers and even different cancer stages can have a certain impact on the analysis results. We also attempted to perform subgroup analysis according to different cancers, but unfortunately, only two trials were on the same cancer. In fact, one trial included breast cancer patients with locally advanced or metastatic disease, while the other included non-metastatic breast cancer. There was still some distinction between the patients in the two trials. This is the limitation of this meta-analysis. We have added this to the discussion.
9) Another point that needs to be more explained is related to the real effect of the use of the ketogenic diet in cancer patients, since, as presented, the main results are associated with anthropometric alteration and not metabolic parameters. In addition, what is the percentage of overweight and obesity in cancer patients? Maybe these data can help the authors to sustain the relevance of the study.
Answer: Thanks for your comment. We are sorry that we don’t have data on individual patients, so we cannot give the percentage of overweight and obesity in cancer patients. But according to the BMI in the baseline characteristics of participants, we inferred that the majority of patients included in the analysis of fat mass change had a BMI >26.8, and some had a BMI >30. The majority are overweight.

Round 2
Reviewer 2 Report
No more comments.